# An Affordable phototherapy intensity meter using machine learning to improve the quality of care system for Hyperbilirubinemia in Indonesia

Yosi Kristian[1], Mahendra Tri Arif Sampurna [2]*, Evan Kusuma Susanto[1], Visuddho Visuddho [3], Kian Djien Liem [4]

1 Department of Informatics, Faculty of Science and Technology, Institut Sains dan Teknologi Terpadu Surabaya, Surabaya, Jawa Timur, Indonesia, 2 Department of Pediatrics, Faculty of Medicine, Universitas Airlangga, Surabaya, Jawa Timur, Indonesia, 3 Medical Program, Faculty of Medicine, Universitas Airlangga, Surabaya, Jawa Timur, Indonesia, 4 Department of Pediatrics, Faculty of Medicine, Radboud University Medical Centre, Nijmegen, The Netherlands

* mahendra.tri@fk.unair.ac.id

**Data Availability Statement:** All dataset for machine learning model in PhotoInMeter are

## Abstract

Hyperbilirubinemia is more frequently seen in low and middle-income countries like Indonesia. One of the contributing factors is a substandard dose of Phototherapy irradiance. This research aims to design a phototherapy intensity meter called PhotoInMeter using readily available low-cost components. PhotoInMeter is designed by using a microcontroller, light sensor, color sensor, and an ND (neutral-density) filter. We use machine learning to create a mathematical model that converts the emission from the color sensor and light sensor into light intensity measurements that are close to Ohmeda Biliblanket's measurements. Our prototype collects sensor reading data and pairs them with Ohmeda Biliblanket Light Meter to create a training set for our machine learning algorithm. We create a multivariate linear regression, random forest, and XGBoost model based on our training set to convert sensor readings to Ohmeda Biliblanket Light Meter measurement. We successfully devised a prototype that costs 20 times less to produce compared to our reference intensity meter while still having high accuracy. Compared to Ohmeda Biliblanket Light Meter, our PhotoInMeter has a Mean Absolute Error (MAE) of 0.83 and achieves more than a 0.99 correlation score in all six different devices for intensity in the range of 0–90 µW/cm2/nm. Our prototypes show consistent reading between PhotoInMeter devices, having an average difference of 0.435 among all six devices.

## Introduction

Hyperbilirubinemia accounts for 40–60% of cases among all hospitalized neonates in the first seven days of life with a risk of bilirubin neurotoxicity [1]. Phototherapy is one method that has been shown to reduce unconjugated bilirubin levels effectively. Currently there are many types of phototherapy equipment commercially available [2]. The clinical response to

available from the Figshare database (https://doi.org/10.6084/m9.figshare.22308550.v1).

**Funding:** This project was supported by a research grant from the Indonesian Directorate General of Higher Education, Research, And Technology, Ministry of Education and Culture under grant PTKN 2022, contract number 159/E5/P6.02.00.PT/2022. The funders had no role in study design, data collection and analysis, decision to publish, or preparation of the manuscript.

**Competing interests:** The authors have declared that no competing interests exist.

phototherapy depends on the effectiveness of the phototherapy device to reduce bilirubin levels as a result of a balance in bilirubin production and elimination. The dose of phototherapy is the light intensity given in measurable doses [3]. In fact, even though in developing countries, the variability of light intensity used for phototherapy is still too low [4].

Many studies have shown that developing countries such as India, Nigeria and Cameroon are reported to have less than optimal phototherapy light intensities below the therapeutic range [5, 6]. In Indonesia, a study of 17 hospitals found that 8 hospitals had phototherapy lamps below the therapeutic value [7]. These results indicate that suboptimal phototherapy is frequently occurring and may even be ineffective in reducing bilirubin levels. Evaluation phototherapy device intensity in level III hospital also showed variable results of wide-ranging intensity from sub-therapeutic dose to super high intensity which may unnecessary and may cause harm [8]. Thus, knowing the level of reduction in the radiation intensity of phototherapy devices is very important to prevent ineffective phototherapy so that it could reduce exchange transfusion (ET) procedure and complications of severe hyperbilirubinemia which can cause brain damage and permanent deafness.

The American Academy of Pediatrics (AAP) recommends measuring the dose intensity of the phototherapy device periodically [2]. However, current practice in Indonesia is to replace phototherapy lamps based on the length of time they are used according to the manufacturer's recommendations. If we do not apply the current recommendations, then the phototherapy is not effective and it can contribute to severe hyperbilirubinemia which leads to increased morbidity and mortality of newborns in Indonesia. Study from [5] reported that such practice of replacing lamp based on solely on the length of time are used can lead to substandard practice of phototherapy.

The fundamental problem of phototherapy in Indonesia is the unavailability of measuring the intensity of phototherapy because it is still relatively expensive to provide in health facilities, so that this research is expected to produce an innovative output in the form of an affordable price of phototherapy intensity measuring device called PhotoInMeter. Machine learning is a subfield of artificial intelligence and computer science which focuses on using data to simulate the way things work. Machine learning can create a mathematical model that converts the emission from the color sensor and light sensor into light intensity measurements. This research aimed to evaluate the accuracy of PhotoInMeter to estimate the irradiance given by phototherapy device.

## Materials and methods

### Electrical component selection

The design process for the prototype starts by choosing sensors and electrical components for the prototype. Sensors are chosen based on read data quality and price of the sensor. First, we pick the best color sensor based on our design constraint. The color sensor is used to measure the intensity of red, green, and blue colored light wave. We tried multiple different color sensors, and after several tries, we found that the TCS34725 color sensor works well enough and have a reasonable cost for mass production.

TCS34725 [9] is a color sensor developed by Adafruit. It has RGB and Clear light sensing elements to measure the color of an object. It is also equipped with Infra-Red (IR) blocking filter which minimizes the IR spectral component of the incoming light and allows color measurements to be made accurately. TCS34725 uses Inter-Integrated Circuit (I2C) protocol to communicate with the microcontroller.

Next, we choose the light sensor for our prototype. The purpose of this light sensor is to measure direct intensity of the phototherapy device and room ambient light intensity. For this,

we tried 3 different light sensors and experimented with using solar cells as an alternative to light sensor. We found after a few trials that the GY-302 light sensor works best and have a price within reasonable margin. We also found that solar cell measurement results are not very reliable and its size is too big to fit in a small device so it was removed from the final prototype design.

GY-302 Digital Light Intensity Sensor Module is a sensor module based on the BH1750 light sensor. BH1750 [10] is a digital Ambient Light Sensor Integrated Circuit (IC) with an I2C bus interface. It can detect a wide range of light intensity ranging from 0 to 65535 lx. GY-302 uses Inter-Integrated Circuit (I2C) protocol to communicate with the microcontroller.

We use the slightly more expensive organic light-emitting diode (OLED) [11, 12] display for the final model's display component. OLEDs are light-emitting diodes (LEDs) in which the emissive electroluminescent layer is a film of organic compound that emits light in response to an electric current. OLEDs are used to create digital displays in devices such as television screens, computer monitors, portable systems such as smartphones and handheld game consoles. A major area of research is the development of white OLED devices for use in solid-state lighting applications. We use OLEDs as the OLED displays has a much smaller size and does not produce dim light that can interfere with the light sensor.

Finally, we used NodeMCU [13–15] for our microcontroller. NodeMCU is an open-source firmware for which open-source prototyping board designs are available. The firmware uses the Lua scripting language. The firmware is based on the eLua project, and built on the Espressif Non-OS SDK for ESP8266. The prototyping hardware typically used is a circuit board functioning as a dual in-line package (DIP) which integrates a USB controller with a smaller surface-mounted board containing the MCU and antenna. The purpose of the microcontroller is to control the other components and calculating the blue light intensity based on sensor readings. The Wi-Fi module in NodeMCU is for communication purposes, so that PhotoInMeter prototype can be controlled from a smartphone and can send sensor readings to the internet data server.

Other important components in our design include a printed circuit board (PCB), ND filter, and a case. We design a PCB board that can be easily mass produced and can easily connect all components and make them fit in a small handheld device. Neutral-density (ND) filter is used to reduce the intensity of light coming directly into both color and light sensors. This is because it was found in several experiments that without ND filter, PhotoInMeter can only measure up to 30 μW/cm2/nm as the sensor cannot read higher light intensity than 30 μW/cm2/nm.

ND filter reduces incoming light intensity which allows PhotoInMeter to measure more than 120 μW/cm2/nm. Finally, we design a 3D printed case to hold all components, ND filter, and a 9-volt battery as the device's power supply. The final design of PhotoInMeter prototype can be seen in Fig 1. We estimated that the cost of producing 1 prototype of PhotoInMeter is 20 times more affordable than buying 1 Ohmeda Biliblanket.

The schematic of our circuit is presented in Fig 2 and the final design of our PCB is presented in Fig 3. We connect the Serial Data (SDA) pin of TCS34725, GY-302, and OLED module to NodeMCU's SDA pin, which is located in D2. We then connect the Serial Clock (SCL) pin of TCS34725, GY-302, and OLED module to NodeMCU's SCL pin, which is located in D1. We connect the SDA and SCL pin of every module to 1 SDA and SCL pin in NodeMCU to conform with the I2C standard (all modules are connected in 1 line and the microcontroller choses which module it communicates with using device address). We then connect the VCC and GND pin of TCS34725, GY-302, and OLED module to the power source. We also connected the VV and GND pin of the NodeMCU to the power source.

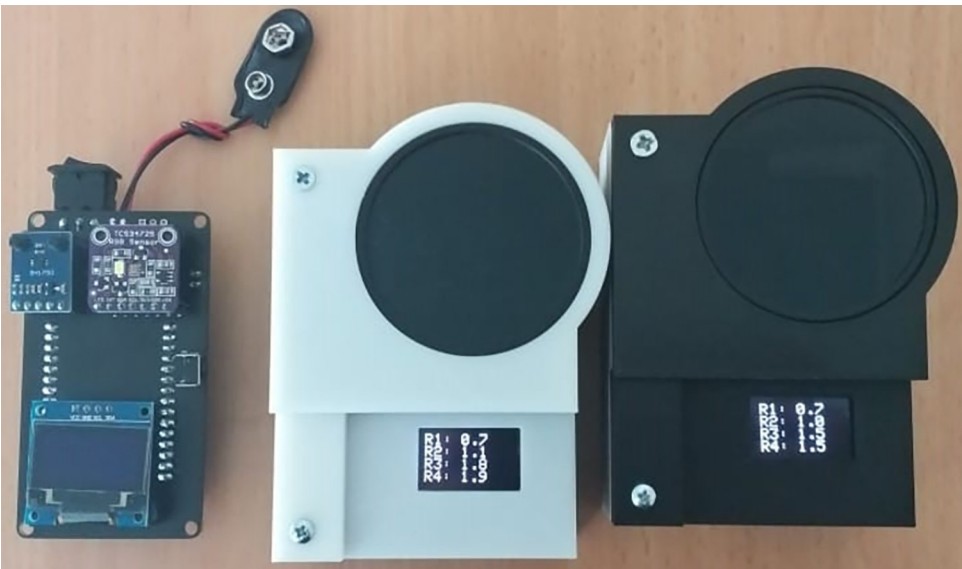

**Fig 1. Final design of PhotoInMeter prototype.**

## Data collection

After we assemble the prototype, we collect our training data to train the mathematical model. Each training data consist of features and a label. We use PhotoInMeter sensors' readings as the feature for our training data. The features list consists of the value read by the light sensor, and 3 values (red, green, and blue) read by the color sensor. We then used Ohmeda Biliblanket Light Meter as the reference measurement device. Ohmeda Biliblanket Light Meter is a standard intensity meter produced by General Electric. It is calibrated annually to maintain the accuracy of the device. We use the measurement provided by Ohmeda Biliblanket as the label for our training data.

We use a silhouette model and multiple different phototherapy devices to help collect our training data. We measure the intensity with the help of the silhouette model which represents 5 points: the head, chest, stomach, legs, and feet of the baby. We repeat each measurement 5 times and use the mean of the values. We use multiple phototherapy devices available in RSUD Dr. Soetomo to collect feature data. The details of each phototherapy device is presented in Table 1 (data taken from [7]). We also use a high intensity white LED lamp to collect data with 0–90 µW/cm2/nm intensity. We use white and light-yellow LED lamp to produce a robust mathematical model that avoids overfitting to blue light data and more resistant to white light noise from the environment.

The method in which we collect data is as follows. First, we put silhouette model under phototherapy device in a certain distance. We then measure each reading from Ohmeda Biliblanket Light Meter in each position marked on silhouette model. After that, we collect sensor readings from each marked position on silhouette model and store PhotoInMeter sensor readings and Ohmeda Biliblanket Light Meter measurement. We repeat this procedure until we have enough data to train a robust mathematical model. In each repetition, we slightly modify silhouette model position and distance relative to phototherapy device so that we have a varying degree of light intensity from 0–90 µW/cm2/nm as training data.

## Mathematical model

The final step in our process is to make a mathematical model using regression to convert sensor readings to accurate measurements. This process consists of 2 steps: preprocessing the raw

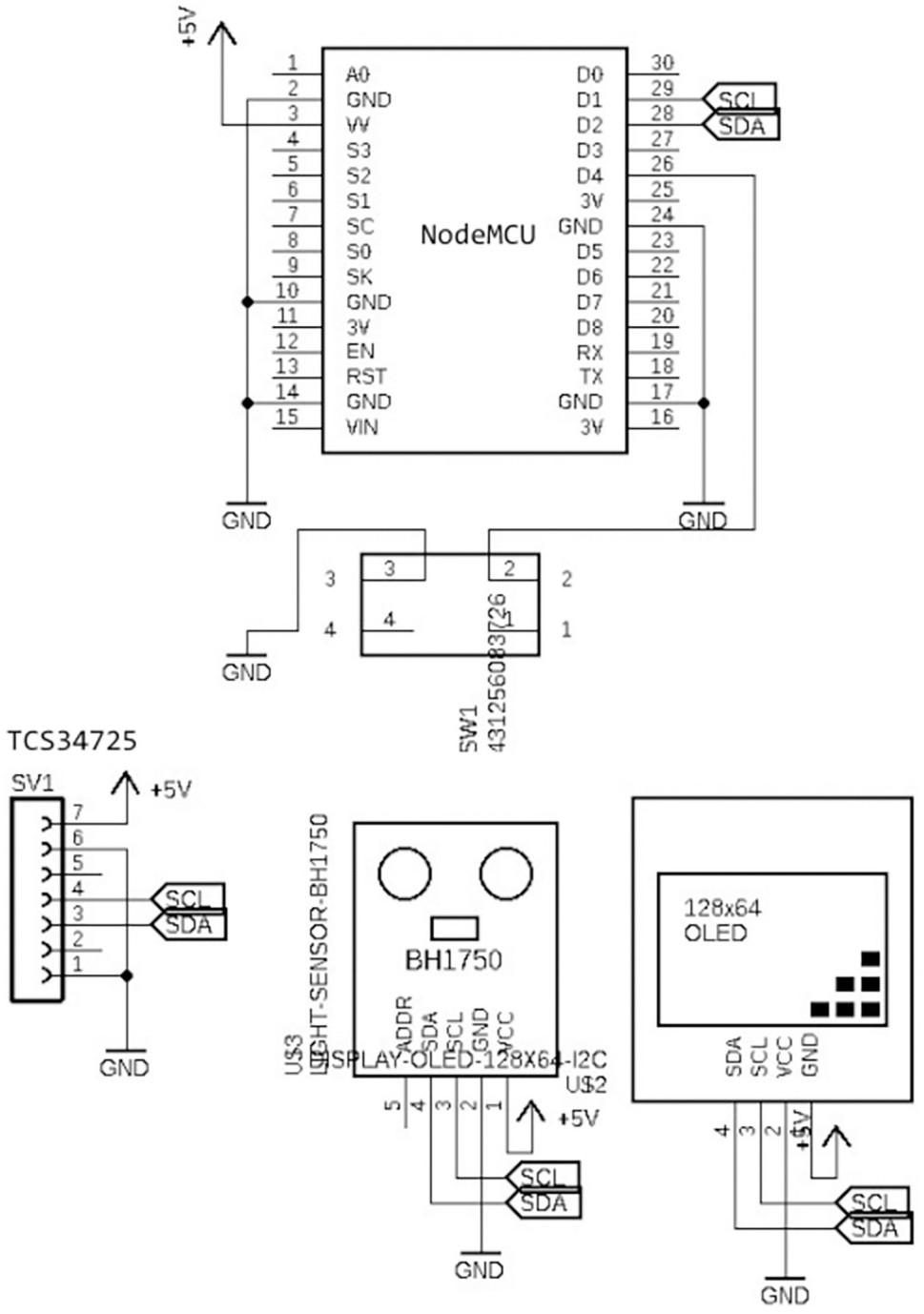

**Fig 2. PhotoInMeter's circuit schematics.**

data and training our models. We tried 3 different machine learning regression models for our experiments. We will then compare each model to see which model will perform best.

Our preprocessing step consists of data normalization and feature construction. We normalize our measurement training data so that our features are now in the range of about 0–1. We then construct additional features (from our 4 features: light, red, green, and blue) by multiplying each feature with each other to make the second-degree polynomial features (light *

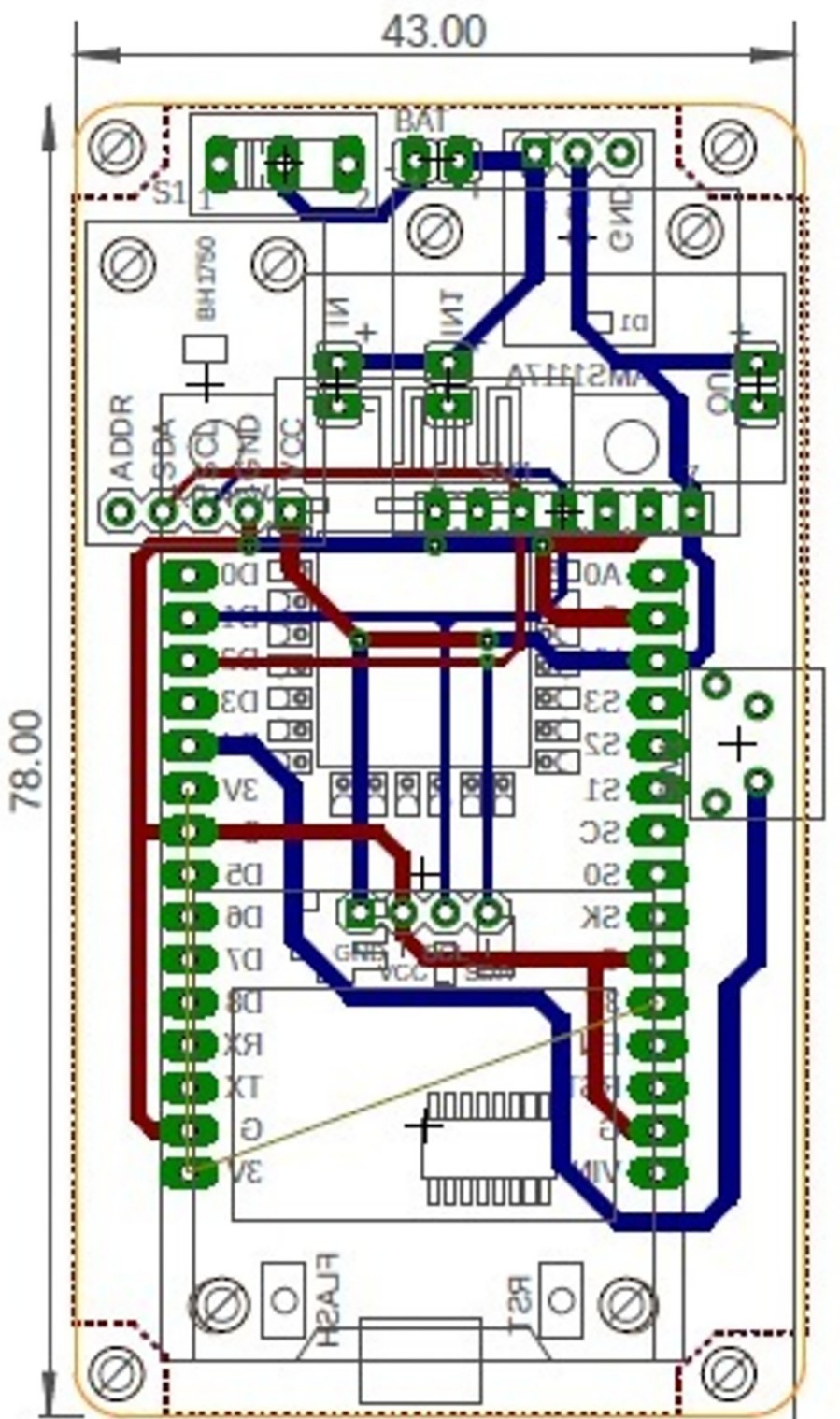

**Fig 3. PhotoInMeter PCB design schematics.**

**Table 1. Phototherapy (PT) device, type, range of irradiance levels, and distances measured.**

| PT Device | Type | Irradiance Levels | Distances Measured |
|---|---|---|---|
| GEA XHZ 90 | Fluorescent | 6.6–39.4 | 25–59 |
| Medela | Fluorescent | 3.5–38.5 | 28–40 |
| Novos Bilisphere LED | LED | 65.3–95.7 | 17 |
| Nidea PT 2000–1600 | Fluorescent | 7.1–16.1 | 30 |
| Seefar 4000 Spot | LED | 44.4–80.6 | 25 |

light, light * red, light * green, light * blue, red * red, and so on). After this step, we now have 14 different features, and each feature will have a value between 0–1. We will now use these features to train our mathematical model.

For the mathematical model, we use a multivariate linear regression [16], a random forest regressor [17], and an XGBoost regressor [18]. We use Lasso regularization with $\alpha = 0.001$ for our multivariate linear regression and we use 20 trees with a maximum depth of 5 for the random forest and XGBoost regressor. We build the mathematical model using Scikit-learn [19] and Orange [20, 21] library and use the preprocessed data in the last section to train the model. This process results in a set of weights (for the multivariate regression) or trees (for the random forest and XGBoost) that can be used to convert sensor reading into blue light intensity measurement.

We also add an extra mathematical model to accommodate the variance in hardware sensitivity. We found throughout our experiments that our hardware has some variance in measurement results for the same reading. Fortunately, calculation results from our models still have a high correlation score compared to Ohmeda Biliblanket Light Meter, which implies that a simple linear regression model can be used to improve the results of the first mathematical model.

The device calibration phase is done by collecting several samples of data from Ohmeda Biliblanket Light Meter and the device to be calibrated. We perform 11 measurements for 11 different light intensities using Ohmeda Biliblanket Light Meter and PhotoInMeter with the 3 trained mathematical models. We then perform a linear regression using these 3 sets of data (Biliblanket and multivariate regression, Biliblanket and random forest, Biliblanket and XGBoost). We then take the coefficient and the intercept from each linear regression result to calibrate our devices' readings. The formula is shown in (1), where f(x) is the output of the first mathematical model, m is the calibration coefficient, c is the calibration intercept, and y is the final reading of the device.

$$y = m \times f(x) + c \tag{1}$$

## Results

We evaluate the performance of our mathematical model in two experimental scenarios. In our first scenario, we compare our model's measurements against those taken using our reference device. This scenario aims to prove that our mathematical model can replicate the behavior of our reference measuring device. For our second scenario, we compare the measurements from multiple prototypes of PhotoInMeter against each other. This scenario aims to prove that our mathematical model can produce similar results even when deployed to different devices.

For the first scenario, we compare the measurements from the PhotoInMeter prototypes against measurements taken using Ohmeda Biliblanket Light Meter. We do this by collecting testing data consisting of the measurements taken with PhotoInMeter prototypes and with

Ohmeda Biliblanket Light Meter. We collect our testing data using the same procedure as collecting our training data. We measure the intensity with the help of the silhouette model to position both devices, repeat each measurement 5 times, and use the mean of the values as measurement data. We use the same phototherapy devices available in RSUD Dr. Soetomo to collect testing data.

For our second scenario, we deploy our mathematical model into six different PhotoInMeter prototypes to test our model. We do this by collecting testing data which consists of the measurements taken with 6 different PhotoInMeter prototypes. We use 6 different prototypes to compare the variance of measurements between different devices. Each prototype uses the same color sensor, light sensor, and ND filter. In total, we collected 11 entries of data for each PhotoInMeter device. The result of our measurements for both scenarios 1 and 2 are presented in Table 2.

We use mean absolute error (MAE) [22], Pearson correlation coefficient [23], and Bland Altman method [24] as evaluation metrics. The formula for both MAE and Pearson correlation is presented in (2) and (3).

$$MAE = \frac{\Sigma_{i=1}^{n}|y_i - x_i|}{n} \tag{2}$$

$$r_{xy} = \frac{\Sigma_{i=1}^{n}(x_i - \bar{x})(y_i - \bar{y})}{\sqrt{\Sigma_{i=1}^{n}(x_i - \bar{x})^2}\sqrt{\Sigma_{i=1}^{n}(y_i - \bar{y})^2}} \tag{3}$$

For our first scenario, we calculate the MAE and Pearson correlation of the PhotoInMeter prototype's measurements with Ohmeda Biliblanket's measurements. We present the measurement results of our six uncalibrated devices in Table. Our best measurements have MAEs of 1.773, 1.436, and 1.473 using linear regression, random forest, and XGBoost model, respectively. Meanwhile, the worst measurements have MAEs of 20.18, 19.14, and 17.83 using linear regression, random forest, and XGBoost model, respectively. However, we can see that even though the worst performing device (Device 6) has a very high MAE, we can see that the Pearson correlation score of all devices borders on 1. We present the measurement results of every device after we calibrate it in Table 3. We can see that the MAE has been reduced significantly (0.591, 1.873, and 2.682 using linear regression, random forest, and XGBoost model, respectively), performing as well as the uncalibrated best device.

Bland Altman Plots to compare the best and worst performing Photoinmeter devices before and after calibration to Ohmeda Biliblanket Light Meter are shown in Figs 4 and 5, respectively. Here we can see that our best and worst performing devices agree with Ohmeda Biliblanket Light Meter measurements. We can also see from Fig 5 that our worst-performing uncalibrated device still agrees with Ohmeda Biliblanket Light Meter measurements and can be fixed using a simple linear regression calibration.

For our second scenario, we calculate the mean absolute difference between every 2 different devices from 6 devices when measuring the same light intensity. We perform this calculation using the calibrated readings. The result of this calculation is presented in Table 4. We can see that our devices have an insignificant mean absolute difference between them. The low mean absolute difference suggests that our devices gave consistent measurements when receiving similar inputs.

## Discussion

We have shown from our experiments that our research has successfully created a measurement device that could accurately and reliably replicate the behavior of Ohmeda Biliblanket

**Table 2. Measurement results from uncalibrated PhotoInMeter devices compared to Ohmeda Biliblanket Light Meter.**

| | Device 1 | | | | Device 2 | | | | Device 3 | | | | Device 4 | | | | Device 5 | | | | Device 6 | | | |
|---|---|---|---|---|---|---|---|---|---|---|---|---|---|---|---|---|---|---|---|---|---|---|---|---|
| | BB | LR | RF | XGB | BB | LR | RF | XGB | BB | LR | RF | XGB | BB | LR | RF | XGB | BB | LR | RF | XGB | BB | LR | RF | XGB |
| | 79.4 | 41.2 | 45.4 | 43.6 | 79.9 | 81 | 78 | 77.8 | 79.9 | 73.6 | 80.1 | 81 | 81.1 | 79.2 | 78.8 | 78.2 | 79.8 | 47.7 | 53.8 | 53 | 79.2 | 39.5 | 42.6 | 45 |
| | 71.1 | 36.9 | 42.5 | 41.5 | 71.7 | 71.5 | 72.7 | 74.6 | 71.7 | 65.6 | 70.1 | 70.4 | 72.9 | 69.9 | 72.8 | 74.6 | 71.8 | 42.1 | 44.9 | 43.4 | 71.2 | 35.2 | 38.6 | 44.7 |
| | 63.3 | 32.8 | 32 | 35.4 | 63.8 | 62.4 | 65.3 | 66.4 | 63.8 | 57.8 | 61.8 | 64.7 | 65.2 | 61.2 | 65.3 | 65.5 | 63.7 | 37 | 42.6 | 44.7 | 63.5 | 31.1 | 31.4 | 33.1 |
| | 55.4 | 28.4 | 31.8 | 32.8 | 55.6 | 53.3 | 55.2 | 55.8 | 55.6 | 49.9 | 55.6 | 56.4 | 56.9 | 52.2 | 55.4 | 55.8 | 55.6 | 31.9 | 31.8 | 34.9 | 55.3 | 27 | 29.5 | 30 |
| | 47.6 | 24.3 | 26 | 27.8 | 47.8 | 44.7 | 47.5 | 48.2 | 47.8 | 42.3 | 42.8 | 44.2 | 48.4 | 44 | 47.5 | 48.7 | 47.4 | 27 | 30.1 | 30 | 47.3 | 23 | 24 | 24.6 |
| | 39.2 | 20.1 | 19.8 | 21.8 | 39.4 | 36.3 | 42.6 | 43.6 | 39.4 | 34.8 | 42.6 | 43.6 | 40.1 | 35.8 | 42.6 | 43.6 | 39.4 | 22.1 | 24 | 24.6 | 39.2 | 18.9 | 19.5 | 21 |
| | 31.6 | 16.1 | 15.6 | 18.2 | 31.6 | 28.6 | 31.4 | 32.8 | 31.6 | 27.7 | 31.8 | 30 | 32 | 28.2 | 31.8 | 32.8 | 31.6 | 17.6 | 19.5 | 20.8 | 31.5 | 15.1 | 15 | 15.9 |
| | 23.5 | 12.1 | 14.3 | 14 | 23.6 | 21 | 19.6 | 21.6 | 23.6 | 20.6 | 19.8 | 21.8 | 23.8 | 20.8 | 19.6 | 21.6 | 23.6 | 13 | 14.3 | 14.2 | 23.5 | 11.3 | 12.5 | 13.1 |
| | 15.8 | 8.3 | 9.2 | 9 | 15.8 | 13.9 | 14.4 | 16.8 | 15.8 | 14 | 14.4 | 16.1 | 16 | 14 | 14.4 | 16.8 | 15.8 | 8.8 | 9.5 | 9.3 | 15.8 | 7.7 | 8.7 | 8.4 |
| | 7.8 | 4.4 | 4 | 4.7 | 7.8 | 7 | 7.5 | 7.6 | 7.8 | 7.1 | 7.5 | 7.6 | 7.9 | 7.1 | 7.5 | 7.6 | 7.8 | 4.6 | 4.1 | 4.8 | 7.8 | 4.1 | 4 | 4.7 |
| | 0 | 0.6 | 2 | 2.3 | 0 | 0.6 | 2 | 2.3 | 0 | 0.6 | 2 | 2.3 | 0 | 0.6 | 2 | 2.3 | 0 | 0.6 | 2 | 2.3 | 0 | 0.6 | 2 | 2.3 |
| **MAE** | | 19.16 | 17.83 | 17.11 | | 1.773 | 1.473 | 1.827 | | 4.018 | 1.791 | 1.645 | | 2.955 | 1.436 | 1.473 | | 16.85 | 14.9 | 14.46 | | 20.18 | 19.14 | 17.83 |
| **Correl** | | 1 | 0.992 | 0.998 | | 0.998 | 0.997 | 0.997 | | 1 | 0.996 | 0.997 | | 0.999 | 0.998 | 0.998 | | 0.999 | 0.994 | 0.994 | | 1 | 0.996 | 0.99 |

BB = Measurement from Ohmeda Biliblanket Light Meter

LR = PhotoInMeter with linear regression model

RF = PhotoInMeter with random forest model

XGB = PhotoInMeter with XGBoost model

**Table 3. Measurement results from calibrated PhotoInMeter devices compared to Ohmeda Biliblanket Light Meter.**

| | Device 1 | | | Device 2 | | | | Device 3 | | | | Device 4 | | | | Device 5 | | | | Device 6 | | | |
|---|---|---|---|---|---|---|---|---|---|---|---|---|---|---|---|---|---|---|---|---|---|---|---|
| BB | LR | RF | XGB | BB | LR | RF | XGB | BB | LR | RF | XGB | BB | LR | RF | XGB | BB | LR | RF | XGB | BB | LR | RF | XGB |
| 79.4 | 79.9 | 80.8 | 77.1 | 79.3 | 81.5 | 77.3 | 76.2 | 79.9 | 81 | 81.2 | 80.9 | 81.1 | 83.4 | 79.7 | 78.5 | 79.8 | 81.6 | 83.5 | 81.8 | 79.2 | 80.3 | 81.3 | 79.5 |
| 71.1 | 71.6 | 75.7 | 73.3 | 71.2 | 72.2 | 72.1 | 73 | 71.7 | 72.3 | 71.1 | 70.3 | 72.9 | 73.8 | 73.6 | 74.9 | 71.8 | 72.1 | 69.9 | 67 | 71.2 | 71.6 | 73.7 | 78.9 |
| 63.3 | 63.6 | 57.1 | 62.3 | 63.5 | 63.2 | 64.8 | 64.9 | 63.8 | 63.7 | 62.8 | 64.6 | 65.2 | 64.8 | 66.1 | 65.6 | 63.7 | 63.5 | 66.4 | 69 | 63.5 | 63.3 | 59.9 | 58.7 |
| 55.4 | 55 | 56.8 | 57.6 | 55.5 | 54.3 | 54.8 | 54.4 | 55.6 | 55.1 | 56.5 | 56.3 | 56.9 | 55.5 | 56.1 | 55.8 | 55.6 | 54.9 | 49.9 | 53.9 | 55.3 | 54.9 | 56.3 | 53.3 |
| 47.6 | 47 | 46.5 | 48.5 | 47.6 | 45.9 | 47.2 | 46.9 | 47.8 | 46.8 | 43.6 | 44.1 | 48.4 | 47.1 | 48.2 | 48.6 | 47.4 | 46.6 | 47.3 | 46.4 | 47.3 | 46.8 | 45.8 | 43.9 |
| 39.2 | 38.8 | 35.5 | 37.7 | 39 | 37.6 | 42.4 | 42.4 | 39.4 | 38.6 | 43.4 | 43.5 | 40.1 | 38.6 | 43.2 | 43.4 | 39.4 | 38.2 | 37.9 | 38.1 | 39.2 | 38.4 | 37.2 | 37.6 |
| 31.6 | 31 | 28.1 | 31.1 | 31.5 | 30.1 | 31.4 | 31.7 | 31.6 | 30.9 | 32.5 | 29.9 | 32 | 30.7 | 32.3 | 32.4 | 31.6 | 30.6 | 31 | 32.2 | 31.5 | 30.7 | 28.6 | 28.7 |
| 23.5 | 23.2 | 25.8 | 23.5 | 23.5 | 22.6 | 19.7 | 20.6 | 23.6 | 23.1 | 20.4 | 21.7 | 23.8 | 23.1 | 20.1 | 21 | 23.6 | 22.8 | 23.1 | 22.1 | 23.5 | 23 | 23.8 | 23.8 |
| 15.8 | 15.8 | 16.8 | 14.5 | 15.7 | 15.6 | 14.6 | 15.9 | 15.8 | 15.9 | 15 | 16 | 16 | 16.1 | 14.8 | 16.1 | 15.8 | 15.7 | 15.7 | 14.6 | 15.8 | 15.7 | 16.5 | 15.6 |
| 7.8 | 8.2 | 7.6 | 6.7 | 7.8 | 8.8 | 7.8 | 6.8 | 7.8 | 8.3 | 8 | 7.5 | 7.9 | 8.9 | 7.9 | 6.8 | 7.8 | 8.6 | 7.5 | 7.6 | 7.8 | 8.3 | 7.5 | 9.2 |
| 0 | 0.8 | 4 | 2.4 | 0 | 2.6 | 2.4 | 1.5 | 0 | 1.2 | 2.5 | 2.2 | 0 | 2.2 | 2.3 | 1.4 | 0 | 1.8 | 4.2 | 3.8 | 0 | 1.2 | 3.7 | 5 |
| MAE | 0.436 | 2.673 | 1.4 | | 1.255 | 1.464 | 1.573 | | 0.645 | 1.782 | 1.636 | MAE | 1.191 | 1.327 | 1.4 | | 0.864 | 1.936 | 2.127 | | 0.591 | 1.873 | 2.682 |

BB = Measurement from Ohmeda Biliblanket Light Meter

LR = PhotoInMeter with linear regression model

RF = PhotoInMeter with random forest model

XGB = PhotoInMeter with XGBoost model

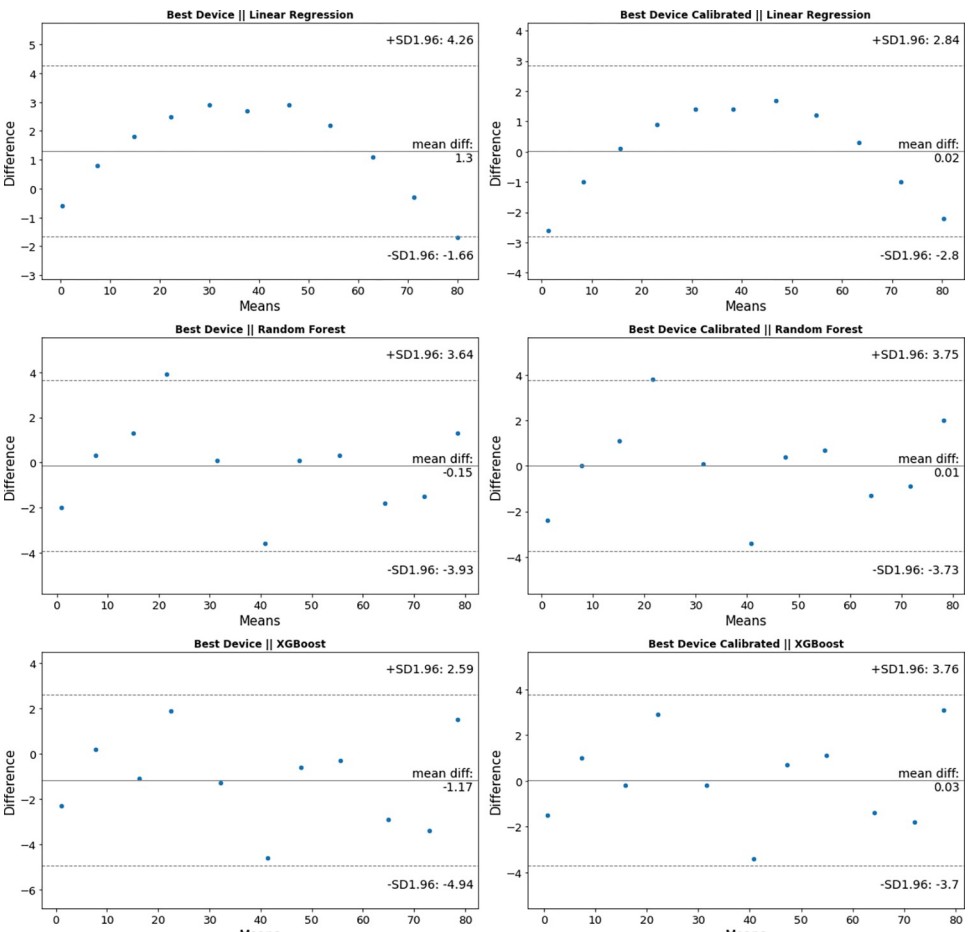

**Fig 4. Bland Altman Plot to compare the best Photoinmeter device before and after calibration to Ohmeda Biliblanket.**

Light Meter. We achieve over than 0.99 correlation score in all six PhotoInMeter prototypes before calibration. After calibration, our devices achieve an average MAE of 0.83, 1.84, and 1.803 for multivariate linear regression, random forest, and XGBoost, respectively. We also show from our experiment that multiple different PhotoInMeter devices have a high consistency. Our evaluation metrics prove that PhotoInMeter has an average reading difference of 0.435 among calibrated devices. These results mean that we can easily mass produce our prototypes without needing many adjustments or calibrations from the hardware perspective. However, we still need to perform calibration from the software side to mitigate the variance in hardware.

From the mathematical model perspective, we found that our best-performing model is made using multivariate linear regression instead of random forest or XGBoost. We suspect that this may mean that there is a simple mathematical formula that can perfectly map simple color sensor readings to light intensity. It is also possible that the random forest and XGBoost method require more training data with varying cases (such as using bright lights with colors other than blue and white). PhotoInMeter also costs 20 times less to produce compared to Ohmeda Biliblanket Light Meter. Using the same amount of money, we can provide 20 different hospitals with a reliable intensity meter. The ability to mass-produce a reliable intensity meter is essential, as many hospitals in Java, Indonesia, do not have an intensity meter [7]. The

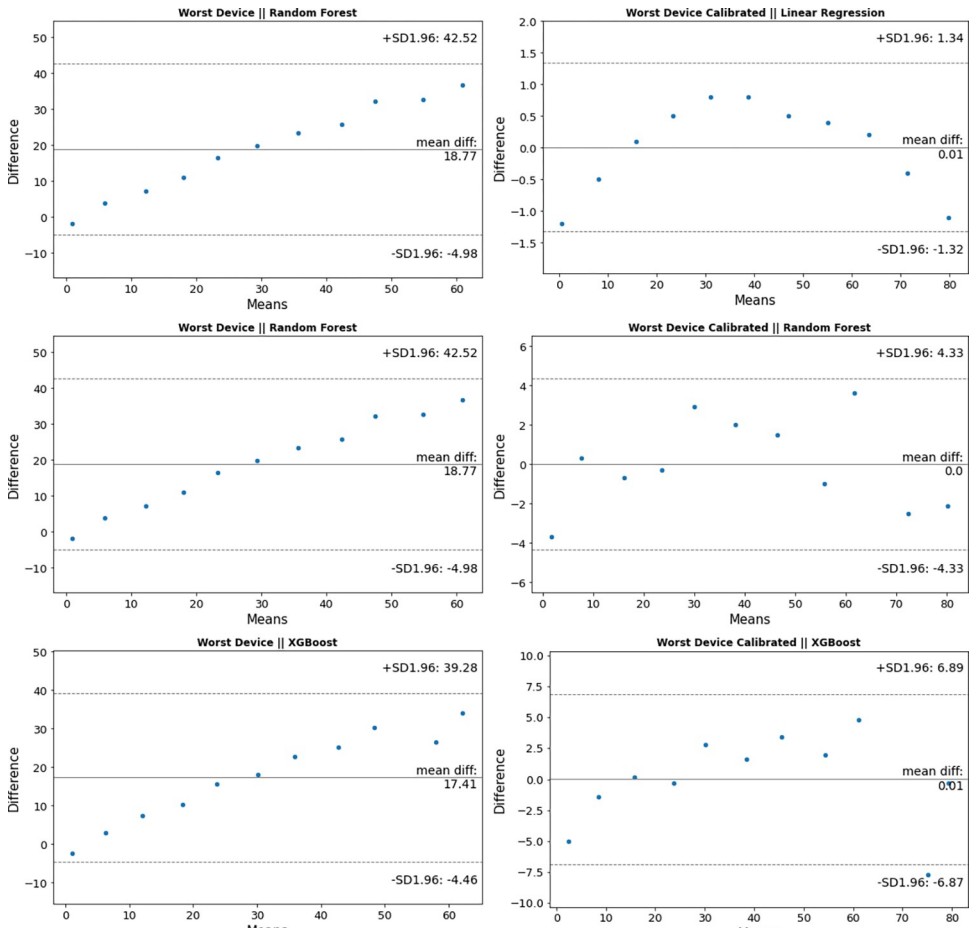

**Fig 5. Bland Altman Plot to compare the worst Photoinmeter device before and after calibration to Ohmeda Biliblanket.**

**Table 4. Mean absolute difference of measurements for each calibrated device compared to each other.**

| First Device Number | Second Device Number | MAE |
|---|---|---|
| 1 | 2 | 0.872 |
| 1 | 3 | 0.263 |
| 1 | 4 | 0.809 |
| 1 | 5 | 0.518 |
| 1 | 6 | 0.245 |
| 2 | 3 | 0.627 |
| 2 | 4 | 0.172 |
| 2 | 5 | 0.390 |
| 2 | 6 | 0.663 |
| 3 | 4 | 0.545 |
| 3 | 5 | 0.327 |
| 3 | 6 | 0.109 |
| 4 | 5 | 0.363 |
| 4 | 6 | 0.618 |
| 5 | 6 | 0.290 |

result of our research may significantly increase the availability of intensity meters in all hospitals and, in turn, increase the quality-of-care.

## Conclusions

This research aims to design an accurate and reliable phototherapy intensity meter with minimal costs. We successfully created PhotoInMeter, an alternative intensity meter that provides accurate reading for low and medium intensity compared to another intensity meter. PhotoInMeter costs 20 times less to produce compared to Ohmeda Biliblanket Light Meter, our reference intensity meter. We achieve more than a 0.99 correlation score in all six PhotoInMeter prototypes. Our best model can achieve 0.83 MAE after calibration compared to Ohmeda Biliblanket Light Meter. PhotoInMeter also has a high consistency among multiple devices, with an average difference of 0.435 among all six devices.

This affordable yet accurate alternative intensity meter can be used to improve the quality-of-care system for Hyperbilirubinemia in developing countries such as Indonesia. In the future, we plan to measure the impact of PhotoInMeter in the quality-of-care system for Hyperbilirubinemia in Indonesia. We will also try to improve the accuracy of higher intensity measurements and define a calibration protocol to mitigate the effects of varying component quality.

## Acknowledgments

We acknowledge Indonesia Ministry of Education and Culture and Ministry of Health who supported the research.

## Author Contributions

**Conceptualization:** Yosi Kristian, Mahendra Tri Arif Sampurna.

**Data curation:** Evan Kusuma Susanto, Visuddho Visuddho.

**Formal analysis:** Mahendra Tri Arif Sampurna, Kian Djien Liem.

**Funding acquisition:** Yosi Kristian, Mahendra Tri Arif Sampurna.

**Investigation:** Yosi Kristian, Mahendra Tri Arif Sampurna, Evan Kusuma Susanto, Visuddho Visuddho.

**Methodology:** Yosi Kristian, Evan Kusuma Susanto.

**Resources:** Mahendra Tri Arif Sampurna, Visuddho Visuddho, Kian Djien Liem.

**Software:** Yosi Kristian, Evan Kusuma Susanto.

**Supervision:** Mahendra Tri Arif Sampurna, Kian Djien Liem.

**Validation:** Evan Kusuma Susanto, Visuddho Visuddho.

**Visualization:** Yosi Kristian, Evan Kusuma Susanto.

**Writing – original draft:** Yosi Kristian, Mahendra Tri Arif Sampurna, Evan Kusuma Susanto, Visuddho Visuddho.

**Writing – review & editing:** Mahendra Tri Arif Sampurna, Evan Kusuma Susanto, Visuddho Visuddho, Kian Djien Liem.

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
