## [Decision Letter · Decision Letter 0]

21 Feb 2023

PONE-D-22-35183An Affordable Phototherapy Intensity Meter Using Machine Learning to Improve the Quality of Care System for Hyperbilirubinemia in IndonesiaPLOS ONE

Dear Dr. Sampurna,

Thank you for submitting your manuscript to PLOS ONE. After careful consideration, we feel that it has merit but does not fully meet PLOS ONE’s publication criteria as it currently stands. Therefore, we invite you to submit a revised version of the manuscript that addresses the points raised during the review process.

We look forward to receiving your revised manuscript.

Kind regards,

Kathiravan Srinivasan

Academic Editor

PLOS ONE

Journal Requirements:

Additional Editor Comments:

Please revise and resubmit your manuscript.

Reviewers' comments:

Reviewer's Responses to Questions

**Comments to the Author**

1. Is the manuscript technically sound, and do the data support the conclusions?

Reviewer #1: Yes

Reviewer #2: Yes

2. Has the statistical analysis been performed appropriately and rigorously? 

Reviewer #1: Yes

Reviewer #2: Yes

3. Have the authors made all data underlying the findings in their manuscript fully available?

Reviewer #1: No

Reviewer #2: Yes

4. Is the manuscript presented in an intelligible fashion and written in standard English?

Reviewer #1: Yes

Reviewer #2: Yes

5. Review Comments to the Author

Reviewer #1: Whether this paper is suitable for this journal can not be judged by this reviewer.

Beside that, this reviewer has only minor comments.

If one reads the abstract only, the role of machine learning is unclear. It only becomes clear in the discussion that the goal is “calibration”. Please provide more information about the actual goal of ML in the Abstract and the Introduction.

Recently in 2022, AAP has revised their almost 20-30 years old guidelines. Please check whether this has an impact on the presented work.

Reviewer #2: The problem of measuring light irradiance in phototherapy devices in low-income countries is important. The paper of Kristian et al. describes a new reliable phototherapy intensity meter which is very cheap in comparison of other meters. The comparison with Ohmeda Biliblanket Light Meter shows a 0.99 correlation score in all prototypes. The paper includes all data concerning the design of prototype and the mathematical model to convert sensor readings to accurate measurements. The paper is interesting and, in my opinion, deserves publication.

6. PLOS authors have the option to publish the peer review history of their article (what does this mean?). If published, this will include your full peer review and any attached files.

Reviewer #1: No

Reviewer #2: No

---

## [Author Response · Author response to Decision Letter 0]

24 Mar 2023

Response to Reviewer

Thank you for your feedback and suggestions on our submission. We would like to thank the reviewer for reading and commenting on our submission. We now have completed and addressed several comments stated in the revisions. Hopefully, revisions made on the manuscript would allow further considerations on indexing the article.

Reviewer #1: Whether this paper is suitable for this journal can not be judged by this reviewer. Beside that, this reviewer has only minor comments.

1) If one reads the abstract only, the role of machine learning is unclear. It only becomes clear in the discussion that the goal is “calibration”. Please provide more information about the actual goal of ML in the Abstract and the Introduction.

Answer : We are very thankful for your comment regarding our manuscript. We now have added the information about ML in the abstract and introduction.

2) Recently in 2022, AAP has revised their almost 20-30 year old guidelines. Please check whether this has an impact on the presented work.

Answer : We are very thankful for your reminder about the new version of AAP guidelines. We have checked, they still have a concern on the measurement intensity of phototherapy. We have renewed our references.

Reviewer #2: The problem of measuring light irradiance in phototherapy devices in low-income countries is important. The paper of Kristian et al. describes a new reliable phototherapy intensity meter which is very cheap in comparison to other meters. The comparison with Ohmeda Biliblanket Light Meter shows a 0.99 correlation score in all prototypes. The paper includes all data concerning the design of the prototype and the mathematical model to convert sensor readings to accurate measurements. The paper is interesting and, in my opinion, deserves publication.

 Answer : We are very thankful for your comment regarding our manuscript.We hope that our manuscript can be beneficial for phototherapy practice especially in low-income countries.

---

## [Decision Letter · Decision Letter 1]

16 Apr 2023

An Affordable Phototherapy Intensity Meter Using Machine Learning to Improve the Quality of Care System for Hyperbilirubinemia in Indonesia

PONE-D-22-35183R1

Dear Dr. Sampurna,

We’re pleased to inform you that your manuscript has been judged scientifically suitable for publication and will be formally accepted for publication once it meets all outstanding technical requirements.

Kind regards,

Kathiravan Srinivasan

Academic Editor

PLOS ONE

Additional Editor Comments (optional):

Reviewers' comments:

Reviewer's Responses to Questions

**Comments to the Author**

1. If the authors have adequately addressed your comments raised in a previous round of review and you feel that this manuscript is now acceptable for publication, you may indicate that here to bypass the “Comments to the Author” section, enter your conflict of interest statement in the “Confidential to Editor” section, and submit your "Accept" recommendation.

Reviewer #1: (No Response)

2. Is the manuscript technically sound, and do the data support the conclusions?

Reviewer #1: Yes

3. Has the statistical analysis been performed appropriately and rigorously? 

Reviewer #1: Yes

4. Have the authors made all data underlying the findings in their manuscript fully available?

Reviewer #1: No

5. Is the manuscript presented in an intelligible fashion and written in standard English?

Reviewer #1: Yes

6. Review Comments to the Author

Reviewer #1: (No Response)

7. PLOS authors have the option to publish the peer review history of their article (what does this mean?). If published, this will include your full peer review and any attached files.

Reviewer #1: No

---

## [Editor Report · Acceptance letter]

24 Apr 2023

PONE-D-22-35183R1 

An Affordable Phototherapy Intensity Meter Using Machine Learning to Improve the Quality of Care System for Hyperbilirubinemia in Indonesia 

Dear Dr. Sampurna:

I'm pleased to inform you that your manuscript has been deemed suitable for publication in PLOS ONE. Congratulations! Your manuscript is now with our production department. 

Kind regards, 

on behalf of

Dr. Kathiravan Srinivasan 

Academic Editor

PLOS ONE